# Addressing the Data Gaps on Child and Adolescent Tuberculosis

**DOI:** 10.3390/pathogens11030352

**Published:** 2022-03-14

**Authors:** Sabine Verkuijl, Moorine Penninah Sekadde, Peter J. Dodd, Moses Arinaitwe, Silvia S. Chiang, Annemieke Brands, Kerri Viney, Charalambos Sismanidis, Helen E. Jenkins

**Affiliations:** 1Global Tuberculosis Programme, World Health Organization, 1211 Geneva, Switzerland; verkuijls@who.int (S.V.); brandsa@who.int (A.B.); vineyk@who.int (K.V.); sismanidisc@who.int (C.S.); 2National Tuberculosis and Leprosy Programme, Kampala 7025, Uganda; moorine.sekadde@gmail.com (M.P.S.); arimo82@gmail.com (M.A.); 3School of Health and Related Research, University of Sheffield, Sheffield S1 4DA, UK; p.j.dodd@sheffield.ac.uk; 4Department of Pediatrics, Warren Alpert Medical School of Brown University, Providence, RI 02903, USA; silvia_chiang@brown.edu; 5Center for International Health Research, Rhode Island Hospital, Providence, RI 02903, USA; 6Department of Biostatistics, Boston University School of Public Health, Boston, MA 02118, USA

**Keywords:** tuberculosis

## Abstract

The burden of tuberculosis (TB) among children and young adolescents (<15 years old) is estimated at 1.1 million; however, only 400,000 are treated for TB, indicating a large gap between the number who are cared for and the number estimated to have TB. Accurate data on the burden of pediatric TB is essential to guide action. Despite several improvements in estimating the burden of pediatric TB in the last decade, as well as enhanced data collection efforts, several data gaps remain, both at the global level, but also at the national level where surveillance systems and collaborative research are critical. In this article, we describe recent advances in data collection and burden estimates for TB among children and adolescents, and the remaining gaps. While data collection continues to improve, burden estimates must evolve in parallel, both in terms of their frequency and the methods used. Currently, at the global level, there is a focus on age-disaggregation of TB notifications, the collection of data on TB-HIV, multi-drug resistant (MDR)-TB and treatment outcomes, as well as estimates of the disease burden. Additional data from national surveillance systems or research projects on TB meningitis, as well as other forms of extra-pulmonary TB, would be useful. We must capitalize on the current momentum in child and adolescent TB to close the remaining data gaps for these age groups to better understand the epidemic and further reduce morbidity and mortality due to TB.

## 1. Introduction

The true burden of tuberculosis (TB) among children and adolescents is not well understood, despite substantial improvements in data collection and availability in recent years and the development of methods to estimate these burdens more accurately. According to the latest World Health Organization (WHO) Global Tuberculosis Report, an estimated 1.1 million children (aged < 10 years old) and young adolescents (aged 10–14 years old) developed TB disease in 2020, and 226,000 died due to TB [1]. Snow et al. have estimated that an additional 586,000 older adolescents (aged 15–19 years old) develop TB each year (797,000 adolescents aged 10–19 years) [2], and researchers have also estimated that approximately 25,000–32,000 children and young adolescents develop multi-drug- (MDR-)TB each year [3,4]. The number of children and adolescents notified to the WHO is substantially lower than these estimates; in 2020, approximately 400,000 children aged 0–14 years were notified with TB [1]. Gaps between notifications to the WHO and burden estimates stem from shortcomings in detection, diagnosis and reporting in these age groups.

The detection, diagnosis, and reporting of children and adolescents with TB is challenging for many reasons, with some of these challenges varying by age. Briefly, for young children, who are more likely to have the paucibacillary disease, current diagnostic tests have low sensitivity to detect *Mycobacterium tuberculosis* [5]. Historically, diagnostic tests for pulmonary TB have relied on having a sputum sample, but young children cannot spontaneously expectorate sputum, and many health care providers do not perform induced sputum or gastric aspirate. More recently, the WHO has recommended an expanded range of specimens to diagnose pulmonary and extra-pulmonary TB, including naso-pharyngeal, gastric aspirate and stool specimens to diagnose pulmonary TB and cerebro-spinal fluid or lymph node aspirate to diagnose extra-pulmonary TB, however, these have not been widely implemented [5,6]. The sensitivity of diagnostic tests using these samples ranges between 46–73%, against a microbiological reference standard (i.e., culture) [5], depending on the specimen and diagnostic test used. 

Although significant advances have been made in the area of TB diagnosis, there are also challenges in identifying children who should be evaluated for TB in the first place. Despite WHO guidance to screen pediatric household contacts of TB patients [7], many countries do not have the resources to consistently carry out contact investigation, resulting in missed opportunities to find and treat children with TB disease or infection. In addition, there is increasing evidence that children become infected outside the home, which makes identifying children who require TB evaluation even more difficult [8]. Although the issues related to diagnostic sensitivity and specimen collection are not as significant by the time adolescence is reached, TB-related stigma and lack of adolescent-friendly health services may cause substantial under-diagnosis in this age group as well [9]. 

In this article, we summarize the improvements in the availability of data on child and adolescent TB in recent years, the development of methods to estimate the burden of childhood TB more accurately, and the remaining data gaps and challenges.

## 2. Availability of Data on the Burden of TB in Children and Adolescents

### Historical Perspective

Following the “Call To Action For Childhood TB” in 2011 [10], the availability and quality of national-level data on child and adolescent TB have improved, and the WHO has published estimates of the burden of TB in children. Figure 1 provides an overview of global milestones related to TB data collection and analysis for children and adolescents. The first TB estimates for children and young adolescents aged below 15 years were included in the 2012 Global Tuberculosis Report [11]. Then, in 2013, the WHO introduced a new recording and reporting framework and TB case definitions, which required age-disaggregated data [12]. The WHO then published guidance for national TB programs on the management of TB in children in 2014 [13], which included a recommendation on reporting TB case notifications in two age groups—for young children aged 0–4 years and for older children and young adolescents aged 5–14 years–as suggested in the original 2006 edition of WHO guidance on childhood TB management [14]. Data on the provision of TB preventive treatment (TPT) in TB household contacts aged below five years have been available since 2015 and have been expanded to include all eligible contacts, in line with the WHO guidelines on TB preventive treatment [15] since 2018. Data on the provision of TPT to people living with HIV have been reported since 2005, including children and adolescents, although not age-disaggregated.

The “Roadmap Towards Ending TB in Children and Adolescents” [16] was launched alongside the UN General Assembly High Level Meeting on the fight against TB (HLM on TB) in 2018. This roadmap provides an agenda for scaling up interventions designed to address TB prevention or management for children and adolescents and highlights the remaining gaps related to data collection, reporting and analysis on TB among children and adolescents, many of which have been addressed since it was published. At the UN HLM on TB, ambitious targets were set to further reduce the burden of TB morbidity and mortality globally, including specific targets for children and young adolescents (<15 years old), described further below. To track progress towards these goals, high-quality data are essential.

In 2019, the WHO requested that countries report additional data on TB in children and adolescents [17]. Countries with electronic case-based reporting systems were requested to report disaggregated data on notifications for more refined age groups: i.e., 0–4, 5–9, 10–14 and 15–19 years, compared with 0–4 and 5–14 years previously. The WHO also asked countries to report overall TB treatment outcomes for those aged 0–14 years as well as the number of children and young adolescents started on MDR/rifampicin resistant (RR-) TB treatment. In addition, data on TB/HIV co-infection in children and young adolescents were requested by WHO [1], in line with the commitments of the Rome 5 Action Plan on Paediatric HIV and TB. The Action Plan aims to support countries to collect and report data on TB treatment initiation and outcomes among children living with HIV [18].

## 3. Data on Child and Adolescent TB Reported to the WHO 

Globally, the number of TB notifications among children and young adolescents aged < 15 years increased from less than 400,000 in 2015 to 523,000 in 2019, largely reflecting improvements in age-disaggregated reporting. In 2020, notifications decreased to just under 400,000 again due to the COVID-19 pandemic [1]. Notifications in children and young adolescents were disproportionately affected by the pandemic, with a decrease in notifications of 28% in children aged under 5 years and 21% in children aged 5–14 years, compared to 18% in people aged 15 years and above. TB treatment coverage (approximated as case notifications divided by estimated incidence, previously referred to as the case detection rate) was 36.5% for children and young adolescents aged < 15 years (down from 44% in 2019) and 27.5% in children <5 years of age in 2020.

In 2020, 101 countries, including 13 of the 30 high TB burden countries (TB HBCs), reported data disaggregated into the four to five-year age categories. Figure 2 shows age-specific notifications for these four age groups for nine TB HBCs that reported in 5-year age groups for both 2019 and 2020. Figure 2 shows that the impact of COVID-19 in these nine countries was most significant in the youngest age groups [19].

Globally, the number of children and young adolescents started on second-line treatment for MDR/RR-TB increased from 3398 to 5586 between 2018 and 2019 and declined again to 3235 in 2020. Overall, only 2.5% of all patients starting second-line treatment were aged < 15 years in 2020.

A total of 127 countries reported the treatment success rate (TSR) among children and young adolescents, including 22 TB HBCs (compared to 19 in 2020). The size of this cohort was almost 390,000 (representing 74% of total notifications in children aged < 15 years in 2019). The overall TSR was 87.6% for the 2019 cohort, ranging from 69% in Papua New Guinea to 100% in Gabon.

Thirty-eight countries reported data on HIV testing and provision of ART in children and young adolescents for 2020, including 16 TB/HIV HBCs, which included 98% of all HIV testing in this age group. The data included the number of children and adolescents with TB with an HIV test result recorded, the number who tested HIV-positive and the number of those with TB/HIV who were on anti-retroviral treatment (ART). Of 210,000 children, 143,000 (68%) with TB had an HIV test result documented. Of these, 7720 tested HIV-positive (positivity rate of 5.4%) and of the children who were found to have HIV infection, 6653 (86%) were receiving ART or were newly initiated on it.

There has been a steady but slow increase in the proportion of eligible child contacts aged below five years receiving TPT, from 7% in 2015 to 35% in 2020. Although there was a slight improvement from 33% to 35% between 2019 and 2020, the estimated number of eligible patients decreased substantially, due to the COVID-19-related decrease in notifications of patients with bacteriologically confirmed TB in 2020. The number of children who received TPT decreased from 433,000 in 2019 to 386,000 in 2020. Fewer than 110,000 contacts over 5 years of age received TPT in 2020.

Details of the data collection and reporting process for one of these HBCs, Uganda, are shown in Box 1 and Table 1, Table 2 and Table 3. The age disaggregation in Table 1 allows a comparison of the proportion of notifications in children across reporting periods to monitor TB case-finding trends. The age-specific treatment outcomes by type of TB and HIV status enable granular analysis and targeted interventions for their improvement (Table 2). Table 3 shows data on the provision of TB preventive treatment to eligible children which aids reporting on the country’s TB prevention efforts and UNHLM targets. Uganda is just one example of a country that has reaped benefits from improving its data reporting and systems for TB among those under 20 years old (Box 1). Globally, in the last four years, work has been carried out to strengthen TB surveillance (under the auspices of the Health Data Collaborative and in collaboration between the WHO, the University of Oslo, and the Global Fund) by transitioning to digital packages (e.g., DHIS2) that make available health facility data and facilitate the analysis (through standardized dashboards) and use of these data to guide policy, planning and programmatic action.

Box 1Case study—Uganda.Uganda is one of the 30 WHO high TB and high TB/HIV burden countries. The country is currently implementing the National TB and Leprosy Strategic Plan 2021/22–24/25 using a person-centered approach. The Strategic Plan highlights the role of high-quality TB data. In 2019, the Ministry of Health initiated national and sub-national multi-stakeholder consultative meetings to review and update the Health Management Information System (HMIS) tools, which were last updated in 2014. The National TB and Leprosy Program (NTLP) was engaged in this process to take stock of data needs and considerations for implementation. Indicators considered for child and adolescent TB were recommended by the pediatric TB committee–a multi-stakeholder platform that provides technical support for child and adolescent TB under the NTLP. The HMIS tools are mainly paper-based and are used to compile routine aggregate reports that are submitted electronically to the national level via the District Health Information System-Version 2 (DHIS2). The new tools were updated to include age-disaggregated data for child and adolescent TB case notifications, treatment outcomes and TB preventive treatment (Table 1, Table 2 and Table 3). Following the finalization of the HMIS tools, a team of national trainers trained regional and district teams who, in turn, trained the health care providers on the new tools and subsequent electronic reporting via the DHIS-2 system. Regular review and data cleaning exercises are conducted at the district and regional levels to ensure data quality. Based on the reported data, the NTLP can assess the ratio between the 0–4 and 5–14 year age groups which assists in identifying age-specific gaps in case finding (see Table 1). With age-disaggregated data, the NTLP is also able to provide burden estimates among the different age categories, as well as compare treatment outcomes for children by HIV status (see Table 2). The more detailed age disaggregates allow the NTLP to track the cascade of TB prevention, screening, diagnosis and treatment among contacts under the age of five years, as well as children and adolescents living with HIV. Currently, Uganda is rolling out a DHIS2 based electronic case-based surveillance system (eCBSS) platform that captures real-time individual-level data which then feeds into periodic reports; this facilitates data utilization from facility to national levels with all the required age categories. The eCBSS is accessible online with offline data entry options for health facilities where there is limited connectivity. The data are displayed in dashboards facilitating point-of-care data use for decision making, including on contact investigation efforts for children and adolescents. It is currently being used in all 17 MDR-TB treatment initiation facilities and an additional 176 health facilities. Scale up to over 1600 health facilities (covering one quarter of all health care facilities in the country) over the next two years is in process.

## 4. Progress towards the UN HLM Meeting Targets for 2018–2022

Current reporting of child and adolescent TB data has allowed assessment of progress towards the UN HLM targets [1]. In the three-year period 2018–2020, 1.44 million TB children and young adolescents were notified of TB by national TB programs; this figure represents 41% of the five-year target of 3.5 million people notified with TB in this age group. In addition, 12,218 children were treated for MDR/RR-TB between 2018 and 2020, which represents only 10.6% of the five-year target of 115,000, indicating that this target is unlikely to be achieved. Between 2018 and 2020, 1.2 million household TB contacts aged under five years received TPT representing 29% of the five-year for this age group for TPT of 4 million. 

### 4.1. Estimates of the Burden of TB in Children and Adolescents

Since the WHO produced the first estimates of TB incidence in children and young adolescents (<15 years old) in 2012, the methods for incidence estimates in this age group have evolved, drawing upon the work of Jenkins et al. [3] and Dodd et al. [4] (described below). 

The current WHO approach stratifies by age and sex [20], across all age groups. Incidence estimates use samples from a Bayesian statistical prior that are consistent with notifications; for children, the Bayesian statistical prior is based on an updated version of Dodd et al. [21] Mortality estimation uses vital registration data and modeling using case-fatality ratios for children from Jenkins et al. [22] in countries lacking vital registration data.

The current WHO TB incidence and mortality estimates include age disaggregated estimates for the 0–4, 5–14, and 15–24 year age groups [1]. In 2020, the WHO estimated that 520,000, 570,000, and 1.6 million people aged 0–4, 5–14 and 15–24 years, respectively, developed TB disease, and among these, 150,000 (29%), 50,000 (9%), and 210,000 (13%) people (without HIV infection) died. 

Other studies that provide estimates of TB incidence in those of age < 15 years have used several approaches, with broadly similar findings to those of the WHO. Jenkins et al. [3] used data on the proportion of children versus adults expected to be sputum smear-positive to adjust age- and sputum smear-stratified notification data. They concluded that around 1 million children and young adolescents aged < 15 years developed TB in 2010. Dodd et al. [21] used a mathematical model of infection and progression from infection to disease to estimate that 650,000 children and young adolescents <15 years developed TB in 22 high-burden countries in 2010, and later extended this analysis to estimate a global incidence of 850,000 in 2014 [4]. More recently, Yerramsetti et al. [23] used a similar approach to Dodd et al. [4], but with calibration to data from 2013 to 2019 in settings considered to have stronger case detection and reporting systems. They estimated TB incidence in children and young adolescents aged < 15 years to be 1 million in 2019. Finally, as part of the Global Burden of Disease (GBD) study, the Institute for Health Metrics and Evaluation (IHME) reports estimates of TB incidence and mortality in five-year age groups [24]. In 2019, they estimated that 790,000 children and young adolescents <15 years developed TB disease.

TB mortality in children and young adolescents < 15 years was estimated by Dodd et al. [25] by applying case-fatality ratios from a systematic review [22]. They estimated 240,000 TB-related deaths among children in 2015 [25], 80% of these in children aged less than five years old, and the vast majority in untreated children. The Maternal and Child Epidemiology Estimation (MCEE) group produces estimates of child and maternal mortality attributable to several causes. Most recently, Perin et al. [26] included TB as a specific cause of mortality. These estimates were based on WHO estimates, with additional age disaggregation applied and further disaggregation into respiratory and non-respiratory causes, based on the WHO mortality database. They estimated 130,000 TB deaths in children aged less than five years in 2019, making TB one of the top ten causes of death in this age group [26]. Based on work undertaken for IHME, Kyu et al. [27] estimated that 66,000 children <15 years died in 2019 (excluding TB deaths among people living with HIV). This lower estimate reflects genuine uncertainty and methodological difference: the GBD approach to mortality is largely based on death data (vital registration, surveillance, or verbal autopsy) which is often absent in settings with high TB incidence and young populations, necessitating statistical extrapolation from other settings.

A number of researchers have used different approaches to estimate TB incidence or mortality among adolescents (aged 10–19 years). In 2018, Snow et al. [28], used notification data in five-year age groups from Brazil, Indonesia, South Africa, Romania and Estonia to estimate global TB incidence for 5–14 and 15–24-year-olds. This approach was updated by Snow et al. [2] to produce five-year estimates in the 0–24 year age range for 2018, estimating that 211,000 and 586,000 people aged 10–14 years old and 15–19 years old, respectively, developed TB disease in 2018. 

In 2021, MCEE estimated TB mortality in the 5–19-year-old population for the first time, in five-year age bands. They reported that 4.8% of the 1.5 million deaths among 5–19 year-olds in 2019 were from TB, equating to 71,000 deaths [29]. Finally, the IHME estimate that 1.8 million people developed incident TB in the 10–24 year age group, and 58,000 died (excluding TB deaths among people living with HIV) in 2019 [24].

### 4.2. Ongoing Data Gaps and How They Might Be Addressed

Despite the recent improvements in data collection and reporting on child and adolescent TB, several gaps remain. Now that more data are being collected in this domain, improving routine data collection and surveillance systems in many countries is important, with a particular focus on data quality and streamlining systems to relieve some of the burden associated with data collection. In addition, linking data systems in primary, secondary, and tertiary care locations, as well as those in any private health system, is critical. Innovative studies, such as inventory studies [30], can help to identify reporting gaps.

For both surveillance and research purposes, there is a fundamental need for clear and consistent use of the age ranges that define younger and older children and young and older adolescents. Children generally were defined as aged 0–14 years, which does not align with the clinical differences in disease manifestation and outcomes within this age range, especially for younger children. The dichotomization of adolescents into children (<15 years) and adults (15 years and over) may also ignore the specific needs and clinical differences of TB in adolescents aged 10–19 years [31]. In addition, there is a lack of standard definitions for how to identify a person with TB (i.e., a “case definition”) within the child/adolescent age range and this is particularly important for younger children given the issues described above and elsewhere with regards to TB diagnosis and reliance on clinical diagnoses. Diagnostic test accuracy and consistent definitions for the diagnosis of TB in children are shortcomings that have implications for accurate data collection.

There are numerous data gaps that could be filled by research or national routine surveillance in settings that have sufficient capacity. For example, although the WHO requests TB notification data by five-year age bands, some countries or researchers might have the capacity to collect data by one-year age bands given the higher risk of TB meningitis, miliary TB, and TB-related mortality [22] in children younger than two years old. In addition, sufficiently detailed data on MDR/RR-TB in children and adolescents are still lacking, specifically in terms of finer age bands and outcome data. The WHO currently requests data on the number of children and young adolescents (0–14 years) who have initiated MDR/RR-TB treatment and the small number reported likely reflects challenges with diagnosis and treatment, as well as data availability. Research groups have compiled an individual patient database to study outcomes for children/adolescents with MDR-/RR-TB [32], and further data to augment this, especially to fill gaps for younger children and those with HIV co-infection would be useful. These types of data would be useful to understand, for example, which regimens are most effective among different age groups and subgroups.

Epidemiological data on TB meningitis (TBM) are also scarce. As a major cause of mortality and long-term sequelae among children [33], more data on the number of children with TBM and their outcomes would help us to understand the burden of TBM, and how we can reduce this. Additional data on other forms of extra-pulmonary TB (EPTB) data would also be useful. Other forms of EPTB are often misdiagnosed, prolonging the duration of illness, increasing the risk of mortality, and limiting a child’s ability to live a normal life. A recent analysis of routinely-collected TB data in Ukraine shed light on the burden and outcomes of EPTB, including predictors [34]. Collection of similar data through research or national surveillance systems where this is possible would provide data to improve the diagnosis and outcomes for children with these forms of TB. 

Inclusion of data on the long-term sequelae of TB can substantially affect assessments of TB burden [35], and those developing TB during childhood or adolescence may contribute disproportionately to life-years lived with disability of reduced health-related quality of life [36]. In particular, TB meningitis leads to neurological sequelae in an estimated 54% of children and adolescents who survive the illness [33]. Osteoarticular TB and pulmonary TB can cause long-term musculoskeletal and respiratory disability, respectively, but the prevalence of these outcomes is unknown. Defining and capturing additional information (through research or national surveillance) on the sequelae of TB disease in these age groups would help quantify the burden of disability or sequelae, which is the first step towards understanding how to treat children and adolescents with TB and how to optimize health systems to provide TB care in the longer term.

Finally, to better understand the success of TB contact management, it will be useful to collect more data on the underlying population at risk of exposure (especially household contacts), those exposed (and thus eligible for preventive treatment), and those who were started on and completed preventive treatment. Yuen et al. have suggested methods to set child-focused TB care targets [37]. We have summarized the main suggestion in Box 2.

In addition, we note that children and young adolescents (<15 years old) are not included in national TB prevalence surveys. Inclusion of these age groups in national TB prevalence surveys is currently not recommended for several reasons including (a) the sample size required to identify enough bacteriologically confirmed children for precise prevalence estimates in this age group, which would be impractical, costly, and logistically challenging, (b) ethical considerations around the mass screening of this age group with radiography, and (c) “over reading” of radiographs. Potentially leading to a substantial number of children being required to provide specimens, which would be uncomfortable and invasive.

### 4.3. Gaps in Burden Estimates and How They Might Be Addressed

While ongoing improvements to data collection and reporting continue, it is essential that burden estimates continue to develop in parallel. Estimates of the burden of infection, disease, and mortality are needed if we are to understand the gaps in detection and treatment. Although two estimates of the burden of MDR-TB in children and young adolescents (<15 years) exist [3,4], they are both based on sparse data and are 8–10 years old, so they are outdated. As data collection in this area improves, more focus on understanding the burden of drug-resistant TB is needed. Again, in line with a large gap in data collection, there are no estimates of the burden of TBM disease, sequelae, or mortality. Given the high mortality associated with TBM and the current lack of estimates, the need for these data is important to understand the scale of the problem and to inform strategies to close the gap between notifications and estimates. 

Many of these gaps could be filled by researchers from modeling groups who have expertise in these areas. Several have already developed methods for other pediatric and adolescent TB burden estimates and we note that multiple methods and estimates, rather than just one, have value in terms of triangulating results and adding to our general understanding of methods and results. We have summarized the main suggestion in Box 2.

Box 2Proposed solutions to the gaps in child and adolescent TB data and estimates.*Globally*:Improved data quality through easy-to-use, efficient, centralized systems with validation checks in place.Standardized definitions of a “person with TB”, especially for younger children.Standardized age definitions of younger/older children and younger/older adolescents.
*Via research projects or national surveillance systems which have capacity:*
TB notification and mortality data by one-year age bands for children under five years old.Tools to collect notification and outcome data for, and development of methods to estimate:MDR/RR-TB, including reporting of case detection and second-line treatment initiation in five-year age groups.Common forms of extra-pulmonary TB, especially TB meningitis.Long-term sequelae due to TB.

## 5. Conclusions

There have been enormous improvements in pediatric and adolescent TB data and estimates over the last decade. These have helped to quantify the extent of the TB epidemic in this age group and plan interventions to address their specific needs, as well as motivating research in this previously neglected age group. We have the opportunity to capitalize on these advances to further improve data collection tools, improve the quality of data, collect additional data where possible and use the data for country-level action to end TB among children and adolescents.

## Figures and Tables

**Figure 1 pathogens-11-00352-f001:**
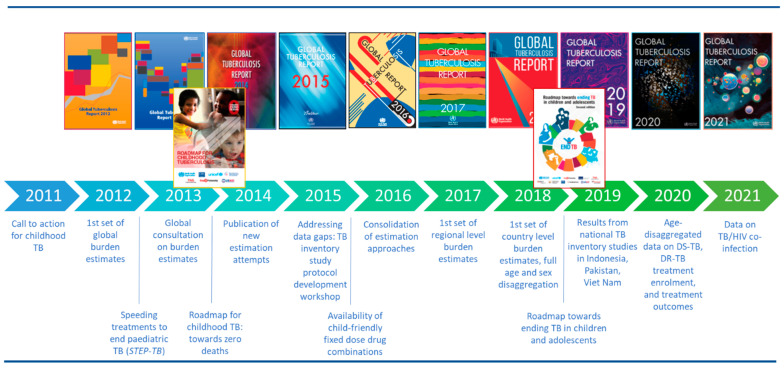
Global milestones related to TB in children and adolescents, 2011–2021 (Reproduced/translated/adapted from “Global TB Report 2021. Geneva: World Health Organization; 2021. Licence: CC BY-NC-SA 3.0 IGO [1]).

**Figure 2 pathogens-11-00352-f002:**
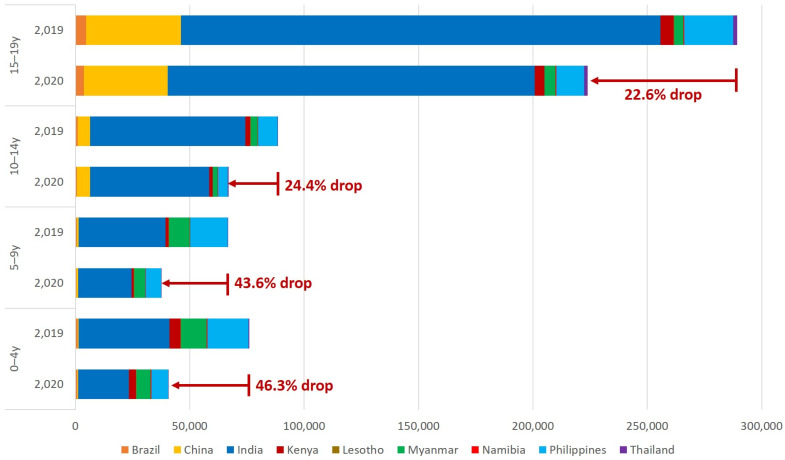
Notifications in five-year age groups for children and adolescents aged 0–19 years in nine TB high burden countries, 2019–2020, with the proportion by which the notifications dropped in 2020 compared to 2019.

**Table 1 pathogens-11-00352-t001:** TB and MDR/RR-TB notifications in Uganda.

		TB Notifications				MDR-/RR-TB Notifications
		Age (Years)				Age (Years)
Time period		0–4	5–9	10–14	15–19	0–14
2020	Jan–Mar	926	520	421	584	11
	Apr–Jun	667	385	335	487	9
	Jul–Sep	885	552	425	514	9
	Oct–Dec	1021	528	410	495	11
2021	Jan–Mar	1279	616	473	590	2
	Apr–Jun	1405	614	498	547	7
	Jul–Sep	1171	611	503	551	2

**Table 2 pathogens-11-00352-t002:** Treatment outcomes for people with TB, TB/HIV co-infection, and MDR-/RR-TB in Uganda among children and young adolescents aged < 15 years old.

TB		Treatment Success (%)	Treatment Failure (%)	Lost to Follow-Up (%)	Died (%)
2020	Jan–Mar	83.7	0.8	10.7	5.4
	Apr–Jun	86.3	0.4	7.6	3.6
	Jul–Sep	84.9	0.1	6.6	4.8
	Oct–Dec	86.5	0.0	6.9	3.9
2021	Jan–Mar	87.3	0.3	4.1	5.1
	Apr–Jun	84.7	0.2	5.0	4.6
	Jul–Sep	89.7	0.2	5.0	3.7
TB/HIV coinfection					
2020	Jan–Mar	80.0	0.6	7.7	7.3
	Apr–Jun	81.6	0.5	7.2	6.0
	Jul–Sep	72.0	0.2	6.3	8.3
	Oct–Dec	90.7	0.0	5.2	7.0
2021	Jan–Mar	81.9	2.2	6.1	10.5
	Apr–Jun	76.7	0.3	3.8	10.8
	Jul–Sep	84.4	0.4	5.2	6.7
MDR/RR-TB					
2020	Jan–Mar	100.0	0.0	0.0	0.0
	Apr–Jun	0.0	100.0	0.0	0.0
	Jul–Sep	50.0	50.0	0.0	0.0
	Oct–Dec	37.5	62.5	0.0	0.0
2021	Jan–Mar	0.0	100.0	0.0	0.0
	Apr–Jun	66.7	33.3	0.0	0.0
	Jul–Sep	0.0	83.3	0.0	16.7

**Table 3 pathogens-11-00352-t003:** Household contacts aged under five years old that initiated TB preventive therapy in Uganda.

Time Period		Contacts Initiating TB Preventive Therapy
2020	Jan–Mar	1009
	Apr–Jun	838
	Jul–Sep	1708
	Oct–Dec	1131
2021	Jan–Mar	1941
	Apr–Jun	2011
	Jul–Sep	2574

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
