# Peer review of "Addressing the Data Gaps on Child and Adolescent Tuberculosis"

_pathogens, 2022, doi:10.3390/pathogens11030352_

Round 1
Reviewer 1 Report
Thank you very much for the opportunity to review this paper on the data gaps for child and adolescent tuberculosis.
The authors did an excellent job of outlining the progress made in terms of paediatric and adolescent TB surveillance data and the methods of disease burden estimation, and I think the paper will make a significant and important contribution to the field.
There are however a couple of key points that I missed in the paper, and I think if included, could further strengthen the paper.
- Currently most TB prevalence surveys do not include children – I think that this is an important remaining data gap that should be mentioned/addressed in this paper.
- I would also like to see a bit more discussion on how improved data could help NTPs to improve services. From a clinical perspective we need actionable data preferably at a facility level, to assist with monitoring and evaluation of TB care to children and adolescents. This is necessary to plan interventions to improve services. TB programs are also moving more and more towards reporting data in a care cascade format – quantifying losses at each step in the cascade can help prioritize urgently needed, targeted interventions to improve care. Having data available in an actionable format, such as a care cascade, can greatly assist NTPs with monitoring and evaluation to improve care.
- Another important challenge from a data perspective, is to quantify the population at risk when reporting on TB contact management. Total number of contacts screened, started on TPT and completed are all important indicators, but to really interpret how well we are doing, we also need to know how many of eligible children (ie TB-exposed children) entered the TPT cascade. In many countries data on household contacts are collected, but this is often not systematically captured or linked to TPT services. Estimates of the population at risk are typically only available at a country level, but there is a need to also have population at risk data available at a lower level.
- In terms of treatment outcomes, mortality and loss-to-follow up are also important considerations in addition to treatment success. Many countries (Indonesia, India, South Africa, Benin) have reported challenges with linkage to care and under-reporting of hospital-diagnosed cases and deaths. This is a particular problem in the paediatric population, seeing as children as often diagnosed at hospital level, but then down-referred to primary health care for continuation of treatment.
- The idea of using Uganda for a practical example, including the case description and data, is good but I don’t think that it is adequately integrated in the manuscript at the moment. If the authors would like to include it, there should be some discussion around what the impact of the changes described in box 1 has been on Uganda’s surveillance data and reporting ability. At the moment I am not sure how it contributes to the paper. There is also no discussion/interpretation of the data that is shown in Table 1 and 2.
- I think measuring our progress towards the UN HLM targets is really important and appreciate that this was included. I would like to know though what the authors think we can do from a data perspective to better support NTPs to reach these targets. For example, strengthening reliability and accuracy of national surveillance systems to minimize diagnosed but unreported cases. I think the inventory studies supported by WHO is a great example of the importance to measure the reporting gap specifically, as both gaps in case detection as well as gaps in reporting/notification impacts on TB treatment coverage
- The authors make a very important point re the importance of including data on the long-term sequelae of TB to measure the burden of TB morbidity. It might be worthwhile to specifically mention the three key areas of post TB health for children and adolescents: TB meningitis (TBM), post-TB lung disease (PTLD) and osteoarticular TB, to align with the WHO operational handbook. These key areas were selected based on the debilitating effects (TBM and osteoarticular TB) or the high incidence of disease and potential long-term impact (PTLD).
Overall, it is an excellent paper, but I think it could be further improved if the authors can highlight how improved data could assist NTPs with monitoring and evaluation of TB care for children and adolescents. Historically, the lack of good surveillance data has limited our ability to respond effectively to the TB epidemic in children and adolescents. But in addition to improved surveillance data, we also need robust monitoring and evaluation strategies to improve care and services for children and adolescents. An example of this is the third of children with TB who doesn’t have an HIV test result recorded. TB is often a missed opportunity to diagnose HIV infection, and all children with TB should have an HIV test. Having data can help us set targets to improve and optimize clinical care.
Minor comments:
- Please add the references for the diagnostic sensitivity range of 46%-73% reported in line 59.
- The following sentence is not clear: In the five-year period 2018-2022, 1.44 million TB children and young adolescents were notified with TB by national TB programmes; this figure represents 41% of the five-year target of 3.5 million people notified with TB in this age group. (line 174-176). I think that the authors meant that in the past 4 years (2018 and 2021), we have reached only 42% of the 5 year target set for paediatric TB case notifications? Please clarify the text.
- Please use the unit of estimate numbers consistently throughout – I found it a bit confusing when in the middle of the paper in the “Estimates of the burden” paragraph estimate numbers are suddenly reported as per “million people”, resulting in decimal numbers (line 194).
Reviewer 2 Report
This is a nice review of the evolving landscape of TB data in children and adolescents. Please see below for some minor suggestions.
Line 59: Please provide a reference for these estimates
Line 65: Please reference the new WHO screening guidelines
Figure 2: To address the impact of COVID-19 as well as age disaggregation could you set this up as three panels? It would be informative to have one for 2019, one for 2020 and then one with the percent difference in notifications in each of these countries by age. This might be a really nice way of showing age disaggregation while addressing the point from the prior paragraph.
Page 141: From the perspective of ease of readership it would be nice if Box 1 came before Table 1
Line 164: Can you clarify whether 86% were on ART at the time of TB diagnosis or this includes the number who were appropriately initiated on ART after TB diagnosis?
Line 194: Consider including %s for mortality numbers listed.
Line 221: Are there any methodological notes that could help reconcile the much lower estimate from IHME for children 0-14 vs. the others which were specifically 0-5?
Line 267: In addition to the IPD is there a reason that the WHO does not request the same age disaggregation for MDR TB? This surprises me and seems like a next step (or maybe I am misinterpreting the text).
